# IP6 is an HIV pocket factor that prevents capsid collapse and promotes DNA synthesis

Donna L Mallery[1], Chantal L Márquez[2,3], William A McEwan[1], Claire F Dickson[1], David A Jacques[2,3], Madhanagopal Anandapadamanaban[1], Katsiaryna Bichel[4], Gregory J Towers[4], Adolfo Saiardi[5], Till Böcking[2,3]*, Leo C James[1]*

[1]Medical Research Council Laboratory of Molecular Biology, Cambridge, United Kingdom; [2]EMBL Australia Node, Single Molecule Science, School of Medical Sciences, University of New South Wales, Sydney, Australia; [3]ARC Centre of Excellence in Advanced Molecular Imaging, School of Medical Sciences, University of New South Wales, Sydney, Australia; [4]Division of Infection and Immunity, University College London, London, United Kingdom; [5]Medical Research Council Laboratory for Molecular Cell Biology, University College London, London, United Kingdom

**Abstract** The HIV capsid is semipermeable and covered in electropositive pores that are essential for viral DNA synthesis and infection. Here, we show that these pores bind the abundant cellular polyanion $IP_6$, transforming viral stability from minutes to hours and allowing newly synthesised DNA to accumulate inside the capsid. An arginine ring within the pore coordinates $IP_6$, which strengthens capsid hexamers by almost 10°C. Single molecule measurements demonstrate that this renders native HIV capsids highly stable and protected from spontaneous collapse. Moreover, encapsidated reverse transcription assays reveal that, once stabilised by $IP_6$, the accumulation of new viral DNA inside the capsid increases >100 fold. Remarkably, isotopic labelling of inositol in virus-producing cells reveals that HIV selectively packages over 300 $IP_6$ molecules per infectious virion. We propose that HIV recruits $IP_6$ to regulate capsid stability and uncoating, analogous to picornavirus pocket factors. HIV-1/$IP_6$/capsid/co-factor/reverse transcription.
DOI: https://doi.org/10.7554/eLife.35335.001

*For correspondence:
till.boecking@unsw.edu.au (TB);
lcj@mrc-lmb.cam.ac.uk (LCJ)

**Competing interests:** The authors declare that no competing interests exist.

## Introduction

Despite a wealth of assays and data, HIV post-entry biology remains poorly understood. Before the virus can integrate its genes into the host genome, it must reverse transcribe its RNA into DNA, travel to the nucleus, remove its protective capsid (uncoating) and target actively transcribing chromatin. However, the relationship between these processes and the order and location in which they occur remain hotly debated. Reverse transcription (RT) is the first enzymatic step in infection and likely occurs as soon as the viral capsid enters the cytosol. Evidence suggests that it precedes all other processes including uncoating and nuclear import. Depletion of dynein (*Fernandez et al., 2015*) and kinesin one heavy chain (KIF5B) (*Lukic et al., 2014*), required for mechanical transport of viral cores to the nucleus, delays uncoating and nuclear entry but does not affect RT. Moreover, none of the capsid cofactors required for nuclear entry or integration are necessary for RT. Depletion of Nup153, Nup358, CPSF6 or TNPO3 inhibits nuclear entry and/or active chromatin targeting but not RT (*Marini et al., 2015*; *Matreyek and Engelman, 2011*; *Matreyek et al., 2013*; *Zhang et al., 2010*; *Sowd et al., 2016*; *Ocwieja et al., 2011*). Capsid mutants at cofactor interfaces largely phenocopy cofactor dependence, reinforcing this notion. Mutants N57A, N74D and P90A have

**eLife digest** Viruses like HIV invade cells and replicate their genome to create new viruses. To hide from components of our immune system that are active inside the cell, HIV uses a protein shell called a capsid, which protects its genome from detection and destruction. However, the capsid faces an engineering challenge beyond those faced by even the most complex man-made structures. This is because the capsid must be strong enough to survive for hours inside the cell but not so strong that it cannot quickly open when the virus needs to release its genome. How this process, called 'uncoating', is achieved is one of the great unanswered questions in HIV biology.

In 2016, researchers made the unexpected discovery that the HIV capsid is decorated with hundreds of pores: one at the center of every subunit from which it is built. Each pore contains a ring of six positively charged amino acids that should destabilize the capsid and cause it to break apart. Yet similar pores are found on a diverse range of viruses.

Mallery et al. – who include several of the researchers involved in the 2016 work – set out to investigate why the HIV capsid contains the positively charged pores. Initial experiments revealed that a molecule called $IP_6$, which is abundant in cells, can bind to the HIV capsid. To do so, six negatively charged phosphate groups in $IP_6$ match up with the six positively charged residues in the pore.

In a related study, Márquez et al. developed a new method that allows the fate of individual capsids to be visualized through time. Here, Mallery et al. use the method to show that $IP_6$ increases how long the capsid remains intact from several minutes to over 10 hours. This allows HIV to copy its genome inside the capsid, meaning it remains protected while the virus prepares to produce new viruses. Mallery et al. also show that HIV packages more than 300 $IP_6$ molecules into itself when it replicates.

Other viruses called picornaviruses use small molecules called pocket factors to stabilize the capsid and to trigger uncoating. Mallery et al. propose that $IP_6$ is an HIV pocket factor. Just as studies of pocket factors have stimulated the development of anti-picornavirus drugs, understanding the role of $IP_6$ may help to develop new treatments for HIV.
DOI: https://doi.org/10.7554/eLife.35335.002

defective integration and/or infectivity in primary human macrophages (*Ambrose et al., 2012*; *Rasaiyaah et al., 2013*; *Schaller et al., 2011*) but can still synthesise DNA.

The critical role the capsid plays in cellular trafficking and nuclear entry strongly suggests that it remains intact in some form until late in post-entry. Inducing premature uncoating in the cell via the restriction factor TRIM5α and the proteasome (*Stremlau et al., 2006*), or the capsid drug PF74, irreversibly blocks RT and inhibits infection (*Price et al., 2014*; *Shi et al., 2011*). Exactly where and when uncoating normally happens and whether this occurs in a single step, or in sequential partial uncoating steps, is unclear. Nevertheless, one implication of RT preceding all other steps is that complete uncoating cannot be immediate. Encapsidated reverse transcription (ERT) assays demonstrate that RT can take place inside purified HIV cores before uncoating when they are supplied with dNTPs (*Warrilow et al., 2007*). Inside the viral capsid, RT enjoys a high enzyme and substrate concentration and synthesised DNA is protected from nucleic acid sensors and nucleases in the cytosol (*Rasaiyaah et al., 2013*; *Lahaye et al., 2013*). While the benefits of remaining encapsidated are clear, HIV capsids are known to be highly fragile in vitro (*Forshey et al., 2002*) and the accumulation of newly synthesised DNA increases their instability (*Rankovic et al., 2017*). In contrast, it takes at least 10 hr for the virus to complete RT (*Butler et al., 2001*) and dock at the nucleus (*Arhel et al., 2007*).

Recently, we reported that HIV capsid hexamers possess an electropositive pore that is essential for RT and infection (*Jacques et al., 2016*). A ring of arginine residues (R18) within the pore avidly binds dNTPs, potentially explaining nucleotide transport into the capsid interior to fuel RT. However, the properties of the R18 pore and its consequences for encapsidated DNA synthesis and capsid stability remain largely unexplored. Crucially, dNTP is not the only polyanion in the cell nor the most abundant. Here, we show that the highly charged polyanion $IP_6$ is specifically incorporated into HIV virions and directly binds the R18 hexamer ring. We demonstrate that $IP_6$ greatly stabilises HIV

cores, limiting their spontaneous disassembly and allowing efficient encapsidated RT. We propose that HIV uses $IP_6$ to regulate capsid uncoating in a manner reminiscent of picornavirus pocket factors.

## Results

### ATP binds the HIV-1 capsid but does not inhibit ERT

Our previous study on HIV capsid pores suggested but did not determine that they are unable to discriminate between ribo and deoxyribo nucleotides. This is an important question because ATP is typically present at cytosolic concentrations > 100 fold in excess of dNTPs and thus a potential competitor for dNTP recruitment by the capsid. We therefore investigated whether ATP competes with dNTPs for binding to HIV capsid hexamers and whether the pore can discriminate between them. We used differential scanning fluorimetry (DSF) to determine if ATP binds hexamers by testing for thermal stabilisation, as was previously observed for dNTPs (*Jacques et al., 2016*). As previously, we used a mutant hexamer protein that is coordinated by disulphide bridges between monomers through the introduction of mutations A14C and E45C and with additional dimerisation mutants W184A and M185A (*Pornillos et al., 2010*). ATP stabilised hexamers with an almost identical increase in melting temperature (Tm) as dATP (*Figure 1a*). Moreover, other triphosphate ligands were also stabilising suggesting a shared mode of binding. Next, we solved a crystal structure of hexamer bound to ATP to determine its mode of binding (*Table 1*). We observed electron density

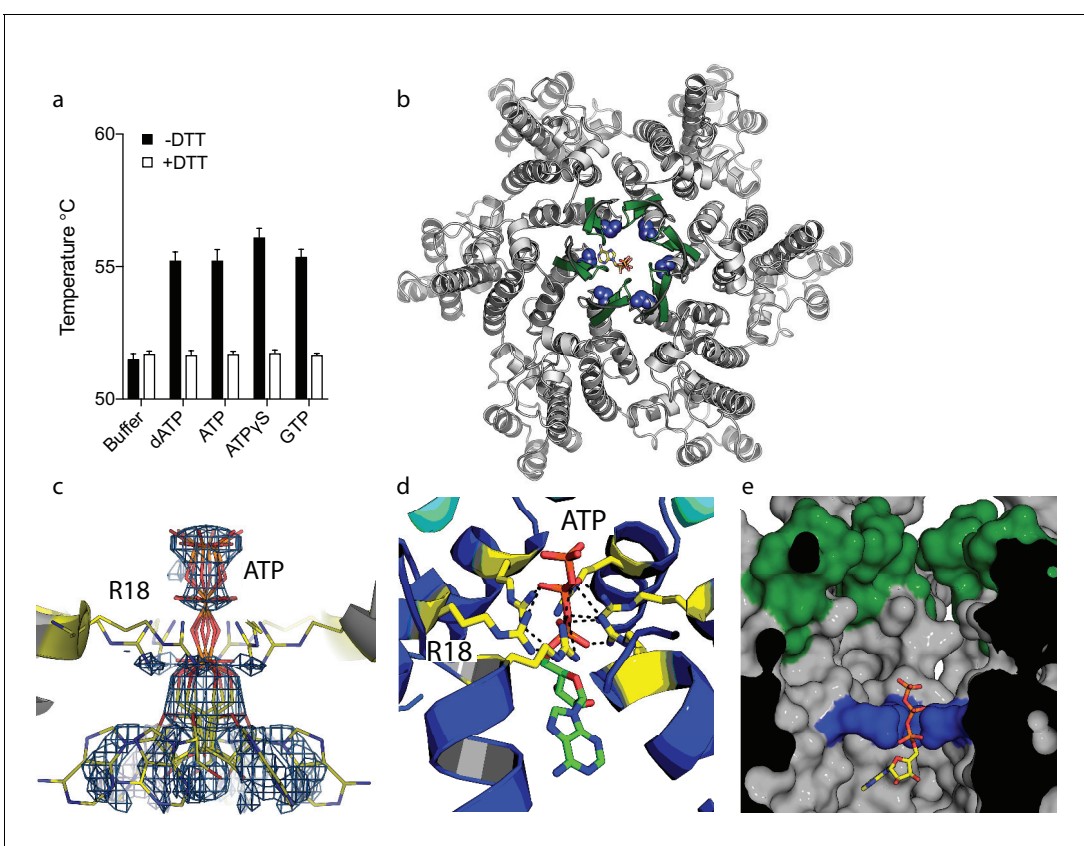

**Figure 1.** ATP binds to the HIV-1 capsid. (**a**) Change in capsid hexamer stability upon addition of different nucleotide triphosphates as measured by differential scanning fluorimetry. Data are from three replicates and representative of at least three independent experiments. (**b**) Complexed structure of ATP bound to HIV capsid hexamer. The N-terminal β-hairpin is coloured in green, the location of R18 in blue spheres and ATP as bonds. (**c**) Electron density for ATP in the complex. (**d**) R18 in the HIV-1 capsid interacts with the adenosine triphosphates. (**e**) A cutaway view of the HIV-1 capsid surface in cross-section showing ATP (as sticks) bound inside the hexamer. The β-hairpin is in green and R18 in blue.
DOI: https://doi.org/10.7554/eLife.35335.003

**Table 1.** Data collection and refinement statistics

|  | 6ERM | 6ERN | 6ES8 |
|---|---|---|---|
| Data collection |  |  |  |
| Space group | P6 | P6 | P6 |
| Cell dimensions |  |  |  |
| $a$, $b$, $c$ (Å) | 91.04, 91.04, 56.53 | 90.85, 90.85, 56.75 | 91.30, 91.30, 57.20 |
| a, b, g (°) | 90.0, 90.0, 120.0 | 90.0, 90.0, 120.0 | 90.0, 90.0, 120.0 |
| Resolution (Å) | 78.85–2.00 (2.03–2.00) | 78.68–2.36 (2.44–2.36) | 79.08–1.90 (2.0–1.9) |
| $R_{meas}$ | 13.3 (59.0) | 7.0 (28.7) | 8.4 (87.8) |
| $CC_{1/2}$ (%) | 99.0 (90.9) | 99.8 (95.0) | 99.1 (94.3) |
| $I$ / $\sigma I$ | 10.8 (6.1) | 15.4 (3.9) | 10.8 (1.6) |
| Completeness (%) | 100.0 (100.0) | 92.6 (92.6) | 91.5 (94.1) |
| Redundancy | 9.3 (9.2) | 4.9 (4.7) | 4.6 (4.4) |
|  |  |  |  |
| Resolution (Å) | 2.0 | 2.36 | 1.90 |
| No. reflections | 18217 | 11142 | 21569 |
| $R_{work}$/$R_{free}$ | 0.23/0.18 | 0.24/0.19 | 0.22/0.18 |
| No. of atoms | 1808 | 1674 | 1816 |
| Protein | 1569 | 1562 | 1623 |
| Ligand/ion | 30 | 30 | 71 |
| Water | 209 | 82 | 122 |
| $B$-factors |  |  |  |
| Protein | 29.0 | 36.87 | 33.2 |
| Ligand/ion | 71.6 | 77.49 | 60.2 |
| Water | 36.3 | 38.39 | 42.7 |
| R.m.s. deviations |  |  |  |
| Bond lengths (Å) | 0.02 | 0.017 | 0.02 |
| Bond angles (°) | 1.90 | 1.82 | 1.90 |

*Values in parentheses are for highest-resolution shell.

DOI: https://doi.org/10.7554/eLife.35335.004

indicating that ATP binds to hexamer at the centre of the sixfold axis and refinement of the six symmetrically equivalent ATP molecules with equal partial occupancy provided a good fit to the data (*Figure 1b,c*). The structure shows that ATP is coordinated solely by the R18 ring confirming that hexamers do not discriminate ribo versus deoxyribo nucleotides (*Figure 1d,e*).

As capsid hexamers do not discriminate NTPs and dNTPs, we tested how the presence of ATP impacts on ERT. ERT reactions were carried out on capsid cores purified from HIV-1 virions by centrifugation through a triton layer and banding in a sucrose gradient (*Shah and Aiken, 2011*). To ensure that this procedure allows recovery of intact cores of the expected morphology, we carried out negative stain electron microscopy (EM) on capsid cores further concentrated by pelleting. We observed intact capsids similar to those described previously (*Welker et al., 2000*) (*Figure 2a*). We investigated RT inside these cores by adding dNTPs, together with DNAse to remove DNA accumulating outside the capsid. Surprisingly, even at a 100-fold excess of ATP over dNTPs, not only did ATP fail to inhibit ERT but it actually increased the quantity of measured transcripts (*Figure 2b*). Similar increases were observed with the non-hydrolysable analogue ATPγS, suggesting that this was not due to an undefined ATPase activity. Improvement in ERT efficiency was dose-dependent, with transcripts increasing over two orders of magnitude as ATP was increased from 10 to 1000 µM (*Figure 2c*). Importantly, ATP had no impact on the activity of purified RT enzyme alone over the same concentration range (*Figure 2d*). Furthermore, substantial ATP stimulation was only observed

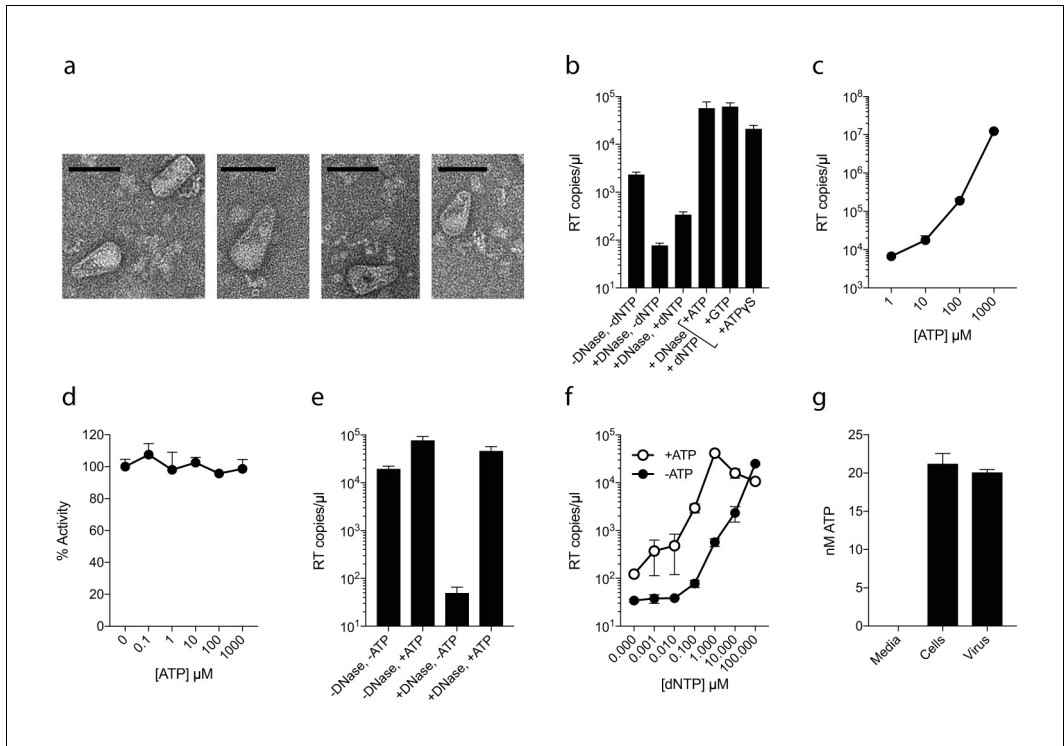

**Figure 2.** ATP does not prevent nucleotide import and ERT. (**a**) Electron micrographs of isolated HIV-1 cores as visualised by negative staining. Size bars are 100 nm. (**b**) ERT in presence of nucleotide triphosphates. (**c**) ATP dose-dependently increases ERT efficiency. (**d**) ATP has no effect on in vitro RT in the absence of a capsid. (**e**) ATP increases accumulation of encapsidated RT products in the presence of DNase. (**f**) ATP increases ERT at dNTP concentrations < 100 µM. (**g**) ATP levels in producer cells (~100 cells) and HIV-1 virions (~5×10$^9$ virions) as detected by luciferase assay (see Materials and methods). All data are mean of three replicates ± SD.
DOI: https://doi.org/10.7554/eLife.35335.005

in ERT reactions carried out in the presence of nuclease (*Figure 2e*). In the presence of DNase, any RT products that are made outside of the capsid, or become exposed as a result of uncoating, will be degraded. Thus, ATP does not alter the efficiency of RT directly but facilitates the accumulation of RT products within intact capsids. This is consistent with the fact that ATP hydrolysis is not required, as shown by the ATPγS data. We hypothesised that the ability of ATP to stabilise hexamers, as measured by DSF, may increase capsid stability during ERT. Consistent with this hypothesis, ATP only promoted ERT at concentrations of dNTPs < 100 µM (*Figure 2f*). At higher dNTP concentrations, ATP presumably had no effect because dNTPs sufficiently stabilised hexamers themselves. This also suggests that different cellular dNTP concentrations (such as during the cell cycle or between different cell types) might alter HIV RT efficiency both directly, by increasing substrate concentration for DNA synthesis, and indirectly by stabilising the capsid. Given that ATP may stabilise hexamers we tested whether ATP is selectively incorporated into HIV particles. ATP was detectable in virions, although at levels approximately similar to those inside the cell (1.5 mM) suggesting no substantial enrichment (*Figure 2g*).

## Non-nucleoside reverse-transcriptase inhibitors (NNRTIs) inhibit encapsidated RT but do not bind capsid pores

The fact that ATP does not inhibit ERT suggests that only a few pores are sufficient for dNTP import or that there are alternative routes into the capsid. To investigate this further, we compared the ability of nucleotide-based reverse transcription inhibitor (NRTI) AZT and non nucleoside RT inhibitiors Rilpivirine (RVP) and Nevirapine (NVP) to block ERT. We hypothesised that if pores represent the only entry route into the capsid then NNRTIs might be less potent in an encapsidated assay because they lack a triphosphate and may fail to interact with the R18 ring. We first used DSF to test

interaction with capsid hexamers and found that only AZT and not the hydrophobic compounds RVP or NVP matched the stabilisation of dATP (*Figure 3a*). A small increase in hexamer stability was observed with RVP but equally in the presence and absence of DTT. This suggests any RVP interaction is not due to the R18 hexameric ring as this feature will be destroyed upon DTT addition because the disulphides holding the hexamer together become reduced. To corroborate these results we measured binding by fluorescence competition with BODIPY-ATP. Only AZT could compete with BODIPY-ATP and was capable of binding capsid hexamers (*Figure 3b*). This data is consistent with pore interaction being dependent on the charged triphosphate group. Determining a structure of capsid hexamer in complex with AZT confirmed that interaction occurs in a similar manner to dATP and ATP (*Table 1*). AZT binds to hexamer at the centre of the six-fold axis and refinement of the six symmetrically equivalent AZT molecules with equal partial occupancy provided a good fit to the electron density (*Figure 3c*). As with the previous complexes, while there is clear density for the phosphates the position of the base can only be inferred. The AZT triphosphates interact directly with R18, which adopts alternative side-chain conformers to allow multiple hydrogen bonds and prevent steric clashes (*Figure 3d*). Taken together with the ATP and dATP structures, this confirms that there are no specific hydrophobic interactions with the pore and is consistent with a lack

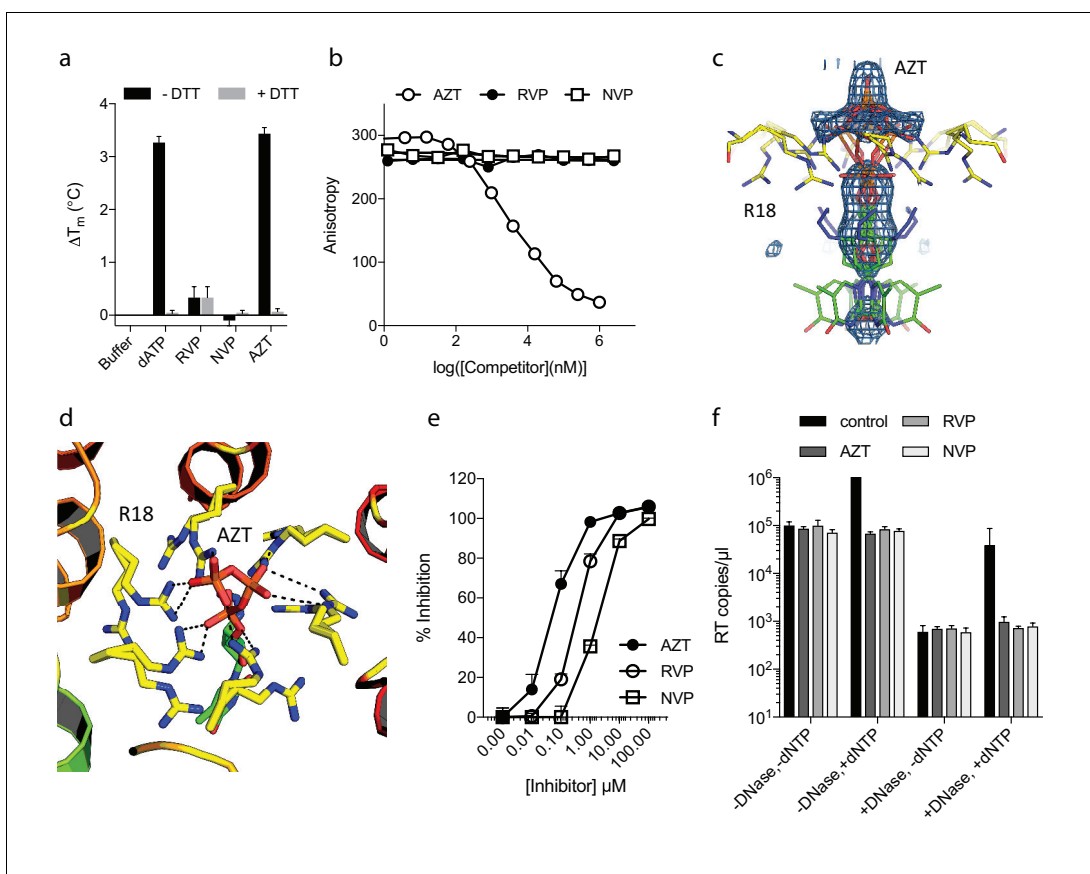

**Figure 3.** NNRTIs and NRTIs are equally potent ERT inhibitors despite differences in pore binding. (a) Changes in hexamer capsid stability upon incubation with dATP or inhibitors in the presence and absence of DTT. Data are averaged from three replicates and representative of three independent experiments. (b) Fluorescence anisotropy competition binding experiments to hexamer capsid with inhibitors. A decrease in anisotropy indicates the ligand has displaced bound BODIPY-ATP. Averaged data from three measurements is shown and is representative of three independent experiments. (c) Electron density for NRTI AZT in complex with HIV capsid hexamer. (d) Putative hydrogen bonds between AZT and capsid reside R18. R18 adopts multiple side-chain conformations. (e) In vitro inhibition of reverse transcriptase enzyme. Data are mean of three replicates ± SD. (f) ERT experiments in presence of inhibitors at in vitro IC90 concentrations. Data are mean of three replicates ± SD.
DOI: https://doi.org/10.7554/eLife.35335.006

of hexamer stabilisation by hydrophobic, uncharged NNRTIs. Next, we compared the activity of NRTI and NNRTIs in RT and ERT reactions to determine whether encapsidation markedly influences inhibitor potency. At the IC90 for in vitro RT inhibition, both AZT and the NNRTIs RVP and NVP blocked ERT with equivalent efficacy (*Figure 3e,f*). The fact that NNRTIs are equally capable of preventing ERT despite showing no interaction with the charged pore suggests that specific binding is not required for entry into the capsid. NNRTIs are sufficiently small to pass through the pore and so may enter by simple diffusion. These experiments also do not rule out entry via a different route entirely.

## Inositol phosphate IP$_6$ stabilises capsid hexamers and promotes DNA accumulation during RT

Our data with ATP suggested that polyanions may have an unappreciated but important role in stabilising HIV capsids and allowing encapsidated DNA synthesis to take place. Previously, we showed that the six-fold symmetric polyaninon hexacarboxybenzene interacts avidly with the R18 charged pore (*Jacques et al., 2016*). Furthermore, it bound hexamers more tightly than dNTP and stabilised hexamers to a greater extent. Hexacarboxybenzene is not a cellular metabolite nor is it bioavailable; however, it is reminiscent of the fully phosphorylated inositol ring of inositol hexakisphosphate (IP$_6$), a strongly negatively charged metabolite possessing six phosphate rather than carboxylic acid groups (*Figure 4a*). Moreover, IP$_6$ has previously been shown to interact with immature capsid and catalyse in vitro assembly of the immature lattice (*Campbell et al., 2001*). We therefore tested by DSF whether IP$_6$ can stabilise the mature capsid and promote ERT by facilitating the accumulation of RT products within the core. To obtain additional information as to stoichiometry of binding, we performed experiments at a range of IP$_6$ concentrations with hexamer protein present at ~33 μM. To show the transition between stabilised species, the melt curves rather than change in stability (ΔT$_m$) are shown (*Figure 4b*). As predicted, IP$_6$ strongly stabilised capsid hexamers, moreover to a substantially greater degree than ATP (*Figure 4c*). A concentration of 20 μM IP$_6$ gave approximately half maximal stabilisation to a solution of 33 μM hexamer. This suggests that hexamer stabilisation by IP$_6$ is likely stoichiometric, with one IP$_6$ molecule per hexamer. Importantly, addition of IP$_6$ to an ERT reaction greatly increased the number of reverse transcripts in the presence of nuclease (*Figure 4d*). This suggests that IP$_6$ promotes the accumulation of RT products within intact capsids. To further test this hypothesis, we repeated our ERT experiment in the presence of the anti-capsid drug PF74. At concentrations > 10 μM, PF74 is thought to destabilise the capsid and induce premature uncoating during infection (*Price et al., 2014*), while in vitro it leads to a rapid failure of capsid integrity (Marquez et al., submitted). We measured no RT products in the presence of both PF74 and nuclease, consistent with induced uncoating and degradation of exposed transcripts (*Figure 4d*).

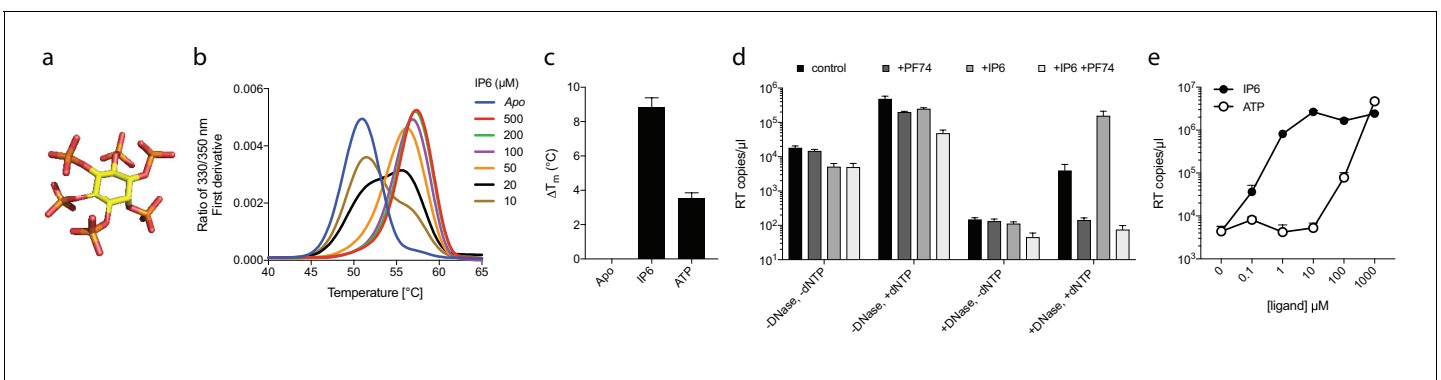

**Figure 4.** Assembly cofactor IP$_6$ promotes ERT. (**a**) IP$_6$ molecule, indicating the non-planar phosphate arrangement. (**b**) IP$_6$ stabilises capsid hexamers with a 1:1 stoichiometry. Data are representative of three different experiments. (**c**) IP$_6$ makes capsid hexamers more stable than ATP. The mean of three measurements ± SD is shown and is representative of three independent experiments. (**d**) IP$_6$ (100 μM) increases RT inside capsid cores (+DNase) but not RT in bulk solution (-DNase). PF74 (30 μM) prevents ERT and counteracts IP$_6$. (**e**) IP$_6$ stimulates ERT more potently than ATP. ERT data are plotted as the mean ± SD of three replicates.
DOI: https://doi.org/10.7554/eLife.35335.007

Importantly, in the presence of PF74, $IP_6$ no longer had any effect. Taken together our observations suggest that $IP_6$ acts to stabilise capsids but that this stabilisation is not sufficient to prevent capsid failure induced by PF74. Nevertheless, in the absence of PF74, $IP_6$ potently stabilised capsid cores at much lower concentrations than ATP. At concentrations where ATP showed no measurable effect, $IP_6$ was able to maximally increase ERT (*Figure 4e*).

## $IP_6$ transforms capsid stability from ∼7 min to >10 hr

Next, we investigated whether $IP_6$ intrinsically stabilises the HIV capsid or just during RT. Capsid cores are known to be unstable and uncoat after isolation. Consistent with this, addition of dNTPs after 24 hr incubation at room temperature resulted in little measurable ERT, in contrast to dNTP addition at time 0 which resulted in effective DNA synthesis (*Figure 5a*). Experiments were carried out in the absence of nuclease, indicating that the absence of measured RT in pre-incubated samples reflects the inefficiency of transcription outside the capsid. Importantly, however, capsid cores that were incubated in the presence of $IP_6$ for 24 hr before the addition of dNTPs supported a similar level of RT as capsids that had not been subjected to temperature-induced uncoating. Together with the previous data, this suggests that $IP_6$ stabilises capsids both prior to and after RT. To investigate this further, we sought to define the intrinsic stability of an HIV capsid and how this is impacted by $IP_6$. To accomplish this, we employed a newly developed single molecule assay described in detail in an accompanying report. In summary, this method involves tethering HIV virions to glass cover slips and permeabilizing their lipid envelope using a pore-forming toxin (*Figure 5b*). Chimeric virions were used in which GFP is cleaved from Gag during maturation of the particle, resulting in GFP molecules packaged within the mature capsid. The uncoating kinetics of individual capsids was then determined by monitoring the loss of GFP signal (*Figure 5c*). Uncoating, here defined as sufficient core opening to allow the escape of GFP molecules, occurs as a single rapid event at the single-particle level (*Figure 5d*). In the absence of IP6, we observed that HIV capsids are highly unstable and rapidly uncoat with a half-life of ∼7 min (*Figure 5d–f*). Remarkably, addition of even 1 μM $IP_6$ was sufficient to dramatically stabilise virions and increase their half-life well beyond an hour (*Figure 5e and f*). Increasing the concentration of $IP_6$ to 10 or 100 μM stabilised capsids so efficiently that too few virions uncoated during the timescale of the experiment for an accurate measurement of the half-life. However, estimates suggest that 10 or 100 μM $IP_6$ stabilises capsid for 5 or 10 hr, respectively.

## $IP_6$ coordinates the R18 ring in HIV hexamer pores

We hypothesised that the remarkable capsid stabilisation achieved by $IP_6$ is likely because the abundant negatively charged groups allow highly efficient coordination and charge neutralisation of the six arginine side-chains in the pore. To investigate this, we solved the structure of hexamer bound to $IP_6$ (*Table 1*). The electron density for the complex confirmed that $IP_6$ binds the R18 ring and six symmetry copies of $IP_6$ at partial occupancy were an excellent fit to the data (*Figure 6a*). This illustrates that $IP_6$ can bind in a number of alternate conformations, with the axial phosphate in different orientations. The N-terminal β-hairpin of the hexamer adopts a 'closed' conformation in the $IP_6$ complex, with $IP_6$ itself located within the central chamber, above the R18 ring (*Figure 6b*). Weak density was observed that may correspond to additional $IP_6$ molecules in the solvent channel below R18 but their inclusion did not improve the fit of the model to the data. Previous hexamer structures have shown that the R18 side-chain can adopt multiple alternative rotamer conformations. Two alternate R18 conformers are necessary in the $IP_6$ complex to prevent clashes with the ligand and maintain interaction. Despite the fact that $IP_6$ is not planar and does not bind parallel to the R18 ring, alternate R18 conformers allow each arginine within the hexamer the potential for hydrogen bond or electrostatic interactions with a phosphate (*Figure 6c,d*). The structure therefore indicates that $IP_6$ is tightly bound by the R18 pore, would effectively neutralise the abundant positive charge and is likely to coordinate and maintain hexamer assembly.

## HIV packages 300 molecules of $IP_6$ per virion

Inositol hexakisphosphate is an abundant polyanion, present in cells at ∼50 μM concentrations (*Veiga et al., 2006*). It is synthesised from Inositol pentakisphosphate ($IP_5$), which can reach similar concentrations. We therefore tested whether $IP_5$ or other precursor molecules are also

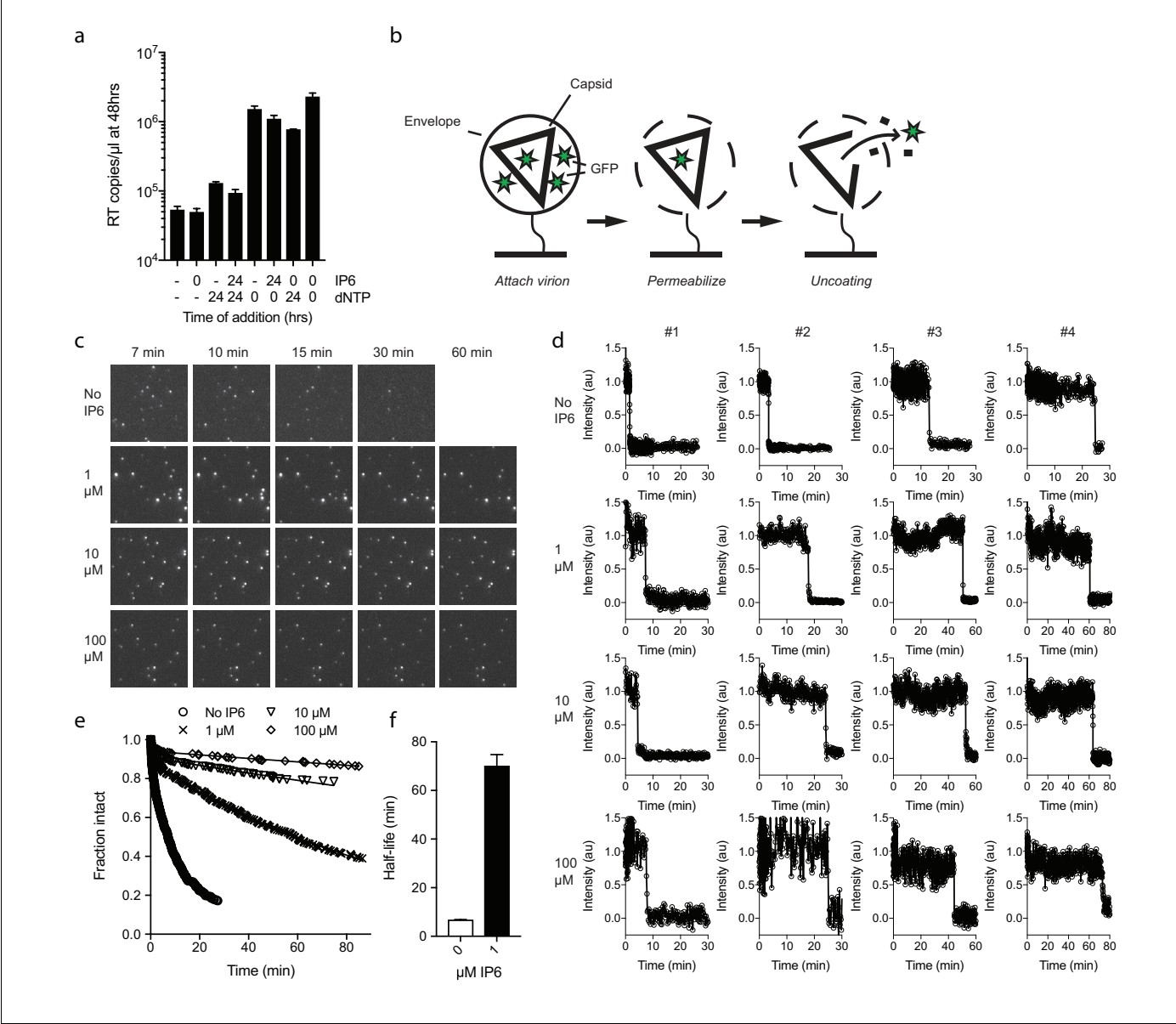

**Figure 5.** IP$_6$ stabilises the HIV capsid both before and during DNA synthesis. (a) IP$_6$, dNTP, both or neither were added to capsid cores at the indicated time points and ERT measured at 48 hr in the absence of nuclease. IP$_6$ added at 0 hr maintains HIV capsid integrity sufficiently to allow ERT when dNTPs are added 24rs later. Data are plotted as the mean ± SD of three replicates. (b) Schematic of the core opening assay. (c) Field-of-view of GFP +intact capsids at different time points and concentrations of IP$_6$ (d) Example traces of single particle GFP release upon core opening. Traces from four individual virions undergoing core opening at different times are shown for each condition. (e) Fraction of intact cores as a function of time (f) Half-life of core opening.

DOI: https://doi.org/10.7554/eLife.35335.008

capable of promoting ERT. DSF showed that other IP compounds, including IP$_5$ and also to a lesser extent IP$_4$, were capable of stabilising capsid hexamers under assay conditions (*Figure 7a*). Addition in ERT experiments at 1 μM concentration revealed a clearer distinction between IP compounds, with IP$_4$ unable to stabilise capsids and promote RT product accumulation in the presence of nuclease (*Figure 7b*). In contrast, IP$_5$ promoted encapsidated RT indistinguishably from IP$_6$. Based on the above data, we hypothesised that IP$_5$ or IP$_6$ could be used by HIV in order to stabilise its capsid. IP$_5$ and IP$_6$ have previously been shown to catalyse the assembly of the immature lattice (*Campbell et al., 2001*), suggesting that these molecules are recruited either during budding,

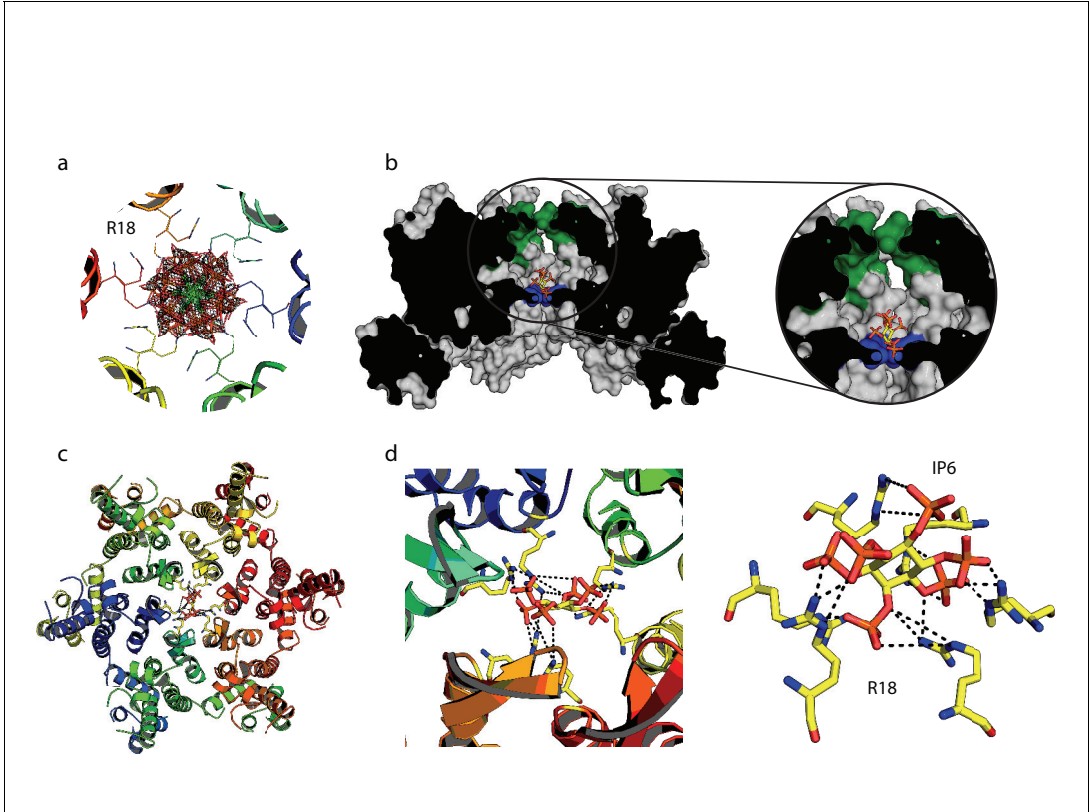

**Figure 6.** IP$_6$ binds the HIV capsid and coordinates all R18 pore guanidino groups. (**a**) View of IP$_6$:hexamer complex down the 6-fold symmetry axis. Secondary structure is colored by monomer. R18 in each monomer adopts two alternate side-chain conformers. All symmetrically equivalent orientations of IP$_6$ present in the structure are shown, together with the 2F$_o$-F$_c$ density (mesh) centred on the ligand and contoured at 1σ. (**b**) View through the centre of the IP$_6$:hexamer complex orthogonal to the 6-fold axis. The molecular surface of the capsid is shown together with a single bound IP$_6$ molecule. N-terminal β-hairpin residues are shown in green and the R18 ring is shown in blue. (**c**) View down the 6-fold axis showing IP$_6$ bound within the capsid hexamer. (**d**) R18 coordination by a single bound IP$_6$ molecule. Putative interactions (2.2–4.2 Å) between each arginine and phosphate is indicated by a black dotted line.

DOI: https://doi.org/10.7554/eLife.35335.009

in a producer cell, or following entry into a target cell. To distinguish between these two possibilities, we attempted to determine whether inositol phosphates are incorporated into HIV virions. We produced viruses from ³H-inositol labelled cells and carefully purified budded viral particles by filtering and multiple centrifugal pelleting steps through sucrose. Virions were acid extracted and strong anion exchange chromatography (Sax-HPLC) was employed to resolve the inositol phosphate present in the virions and in the correspondent infected cell extracts. Scintillation counting of Sax-HPLC eluted fractions revealed the specific presence of IP$_6$ in HIV virions (*Figure 7c*). Importantly, the levels of IP$_6$ were enriched over those of IP$_5$; indeed little IP$_5$ was detected despite the fact that it is present within cells at concentrations around 25% that of IP$_6$. This suggests that it is IP$_6$ selectively incorporated into virions rather than from potential co-purifying microvesicles that is being measured (*Bess et al., 1997*). Subtilisin treatment is used to reduce microvesicle contamination in HIV-1 virion preparations (*Ott, 2009*). To ensure that IP$_6$ detection is not from contaminating microvesicles, we repeated our experiments on subtilisin-treated samples. Subtilisin removed protein contaminants and abolished HIV-1 infectivity consistent with the digestion of envelope proteins, but did not reduce the levels of capsid protein or incorporated IP$_6$ (*Figure 7d* and *Figure 7—figure supplement 1*). Finally, to directly demonstrate that IP$_6$ is associated with the capsid core rather than the viral membrane, we stripped the envelope from produced virions using the same triton-layer centrifugation method as employed to prepare cores for ERT (*Shah and Aiken, 2011*). Incorporation in capsid cores was compared to virions by calculating the scintillation counts for IP$_6$ per μg p24, as determined by ELISA. To confirm that our core prep has removed enveloped particles, we also

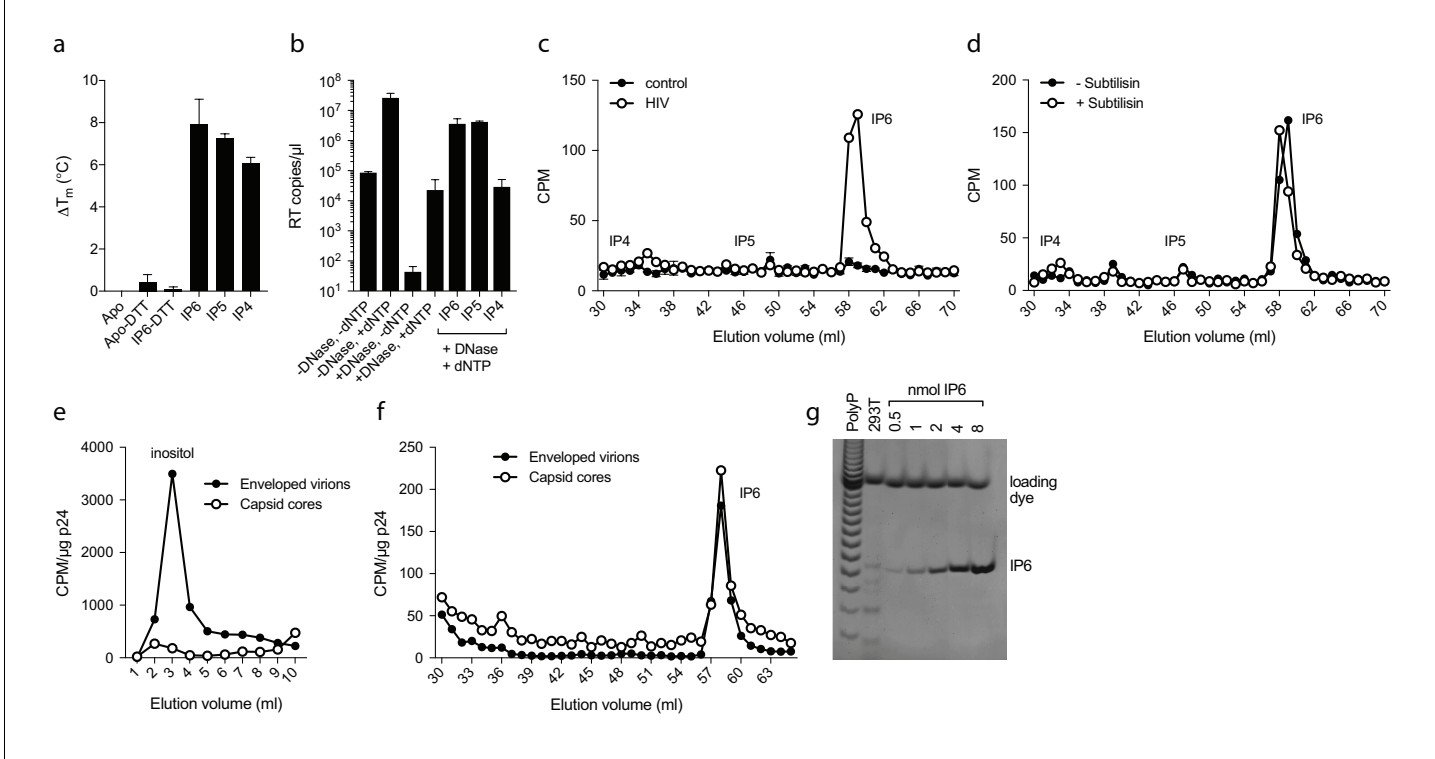

**Figure 7.** $IP_6$ is specifically incorporated into the capsid of HIV virions. (**a**) Hexamer stability in presence of 0.2 µM $IP_4 – IP_6$. Data are averaged from three replicates with SD and are representative of at least three independent experiments. (**b**) $IP_5$ (1 µM) and $IP_6$ (1 µM) promote RT accumulation in presence of nuclease. Data are plotted as the mean of three replicates with SD. (**c**) HIV produced in $^3$H-inositol-treated cells, acid-extracted and Sax-HPLC fractioned reveals specific packaging of $IP_6$ but not $IP_4$ or $IP_5$ into virions. Supernatant of mock transfected cells processed in parallel with viral samples was used as a control. Scintillation data are single measurements representative of three independent experiments. (**d**) $IP_6$ incorporation in virions treated with subtilisin to remove microvesicle contamination (***Figure 7—figure supplement 1***). (**e**) Comparison of inositol levels in samples of intact enveloped virions or purified capsid cores (measured as CPM per µg of p24 capsid). (**f**) As in (**e**) but comparing the levels of $IP_6$. Enveloped virions and capsid cores have equal $IP_6$ incorporation. (**g**) Toluidine blue PAGE of $IP_6$ purified from 293 T cells with known $IP_6$ and polyphosphate (PolyP) standards.

DOI: https://doi.org/10.7554/eLife.35335.010

The following figure supplement is available for figure 7:

**Figure supplement 1.** Subtilisin treatment of HIV virions.

DOI: https://doi.org/10.7554/eLife.35335.011

compared the relative levels of inositol. Inositol is an osmolyte found in the cytosol at ~mM concentrations and is therefore only expected in samples derived from enveloped virions and microvesicles. Consistent with this, we observed loss of the inositol peak in the capsid only sample (***Figure 7e***). However, we observed the same levels of $IP_6$ in both samples demonstrating that $IP_6$ is associated with the capsid core (***Figure 7f***). This result reinforces that measured $IP_6$ is not derived from microvesicles but is enriched into HIV-1 virions and bound to the capsid core. We calculated the number of $IP_6$ molecules per virion by determining the fraction of cellular $IP_6$ that is incorporated. Parallel analysis of the cell extracts by PAGE-based separation revealed that 0.8% of total $IP_6$ radioactivity was incorporated into the produced virions (***Figure 7g***) (***Losito et al., 2009***; ***Wilson et al., 2015***). This corresponds to approximately 309 ± 41 $IP_6$ molecules per viral particle, which is sufficient for 1:1 stoichiometry with hexamer pores present in an intact virion.

## Discussion

Here, we have shown that HIV virions package $IP_6$ when they bud from a producer cell, that $IP_6$ coordinates the electrostatic pores in HIV hexamers and that $IP_6$ dramatically increases capsid stability both intrinsically and during ERT. The logic for requiring a compound such as $IP_6$ to stabilise the

capsid is in part due to the presence of an otherwise repulsive charged structural feature (R18 in the mature capsid). However, if nullifying charge were the only reason for IP6 requirement, then it would be simpler to evolve these residues into apolar alternatives. Thus, using IP6 as a stabilising agent must confer benefits onto the HIV capsid that cannot be achieved by encoding stability via capsid residues alone. A central coordinating ligand like $IP_6$ allows proteins to be stabilised independently of a hydrophobic core or surface and has been used extensively in evolution, giving rise to motifs like the zinc-finger. Small molecule ligands are also highly effective at catalysing multimerisation, something of particular advantage to an assembling viral capsid. Examples where $IP_6$ has been exploited as a small molecule chaperone include HDAC co-repressor complexes, where it promotes stability (*Watson et al., 2016*), and Bruton's tyrosine kinase, where it induces transient dimerisation of the PH-TH module (*Wang et al., 2015*). A further attractive feature of using a small molecule ligand like $IP_6$ to drive self-assembly is that it provides an inbuilt mechanism of disassembly: ligand binding is reversible and so its dissociation can be used as a disassembly trigger.

Using $IP_6$ to stabilise the capsid therefore avoids a fundamental problem in capsid design, namely that the capsid must be strong enough to protect its cargo but not too strong that it can never be uncoated. Based on our data, we propose that $IP_6$ is the HIV equivalent of a picornavirus pocket factor, providing both stability and a mechanism to initiate uncoating. Pocket factors were originally described by Michael Rossmann, who observed a small molecule bound to the picornavirus capsid that he proposed stabilised it (*Hogle et al., 1985*; *Rossmann et al., 1985*). Pocket factors have subsequently been shown to stabilise picornavirus capsids until conformational changes induced by environmental cues during infection promote their dissociation and induce disassembly. We hypothesise that $IP_6$ stabilises the HIV capsid in the cytosol, until its release triggers uncoating. Known capsid cofactors could provide the cues to induce a conformational change and provoke dissociation of $IP_6$. The dynamic N-terminal β-hairpin present in hexamers (*Jacques et al., 2016*) provides one possible mechanism; when closed $IP_6$ will be retained in the pore but when open it would be free to dissociate. Testing this mechanism will be a key goal of future study, as will determining the importance of $IP_6$ in post-entry HIV-1 infection.

While we propose that $IP_6$ serves as an HIV pocket factor, we cannot rule out that other cellular polyanions also play a role. For instance, although we observed no evidence for enrichment, ATP is highly abundant and was readily detected in virions. In our original discovery of the R18 pore and its importance in HIV RT and infection we postulated that it provided a channel to import nucleotides to drive ERT. The cellular data presented in that work is also consistent with the R18 pore being required to bind $IP_6$ in order to stabilise the capsid during its transit through the cytosol. Further experiments are needed to dissect the relative importance of the R18 pore in attracting nucleotides for RT versus binding $IP_6$ for capsid stabilisation. The fact that high concentrations of either ATP or $IP_6$, which would be competitors with dNTPs for pore binding, increase rather than decrease ERT efficiency argues against the R18 pore being essential for nucleotide import. However, as there is a multi-log increase in accumulated viral DNA as a result of capsid stabilisation by polyanions this could mask any decline due to reduced nucleotide import. Capsid permeability is not well understood, but the experiments presented here comparing inhibition by NRTIs and NNRTIs in ERT versus cellular infection suggest that either their efficiency of entry into the capsid is similar – and that interaction with the R18 ring is therefore not essential - or that this is not a limiting step. Either way, this indicates that small molecules not bound by the R18 ring can still enter the capsid through the pore, or through solvent channels as has previously been proposed.

Importantly, as mature hexamers form only after budding, they cannot be responsible for recruiting polyanions such as $IP_6$ into the virion. Of note, $IP_6$ has previously been shown to catalyse immature lattice assembly (*Campbell et al., 2001*), suggesting it has a role in the formation and maturation of capsid during viral production. In the immature hexamer, there are two lysine rings (K227 and K158; equivalent to Gag residues 359 and 290) that are analogous to the R18 ring we have described in the mature hexamer (*Wagner et al., 2016*). We speculate that it is these lysine rings that are responsible for the specific recruitment of $IP_6$ into virions, where it drives immature lattice assembly. Upon protease cleavage and maturation, the six-helix bundle and lysine rings will be lost, making $IP_6$ available to bind the R18 pore in mature hexamers and promote mature lattice formation. Thus, we propose that HIV capitalises on a readily available cellular polyanion to catalyse the assembly of immature and mature lattices, promote their stability and regulate the uncoating of capsid in newly infected cells. The identification of the inositol phosphate $IP_6$ in HIV virions and its role

in imparting onto the capsid a property of metastability provides a new dependence to be exploited in antiviral development.

## Materials and methods

### Protein production and purification

The CA N-terminal domain and the disulfide-stabilised CA hexamer were expressed and purified as previously described, briefly: p24 capsid protein was disassembled in Tris (pH 8.0, 50 mM), NaCl (40 mM), 2-mercaptoethanol (20 mM), then reassembled in Tris (pH 8.0, 50 mM), NaCl (1 M), 2-mercaptoethanol (20 mM). This was followed by oxidation in Tris (pH 8.0, 50 mM), NaCl (1 M) and redispersion in Tris (pH 8.0, 20 mM), NaCl (40 mM). Reassembled hexamers were observed by non-reducing SDS–PAGE.

### Differential scanning fluorimetry

DSF measurements were performed using a Prometheus NT.48 (NanoTemper Technologies) over a temperature range of 20–95°C using a ramp rate of 2.0°C / min. CA hexamer samples were prepared at a final concentration of 200 µM monomer in Intracellular Buffer in the presence or absence of 4 mM DTT. dNTPs or competitors were added at 200 µM. DSF scans are single reads of three replicates and were performed at least three times unless otherwise stated. Consistency between like points yields an uncertainty in $T$m of no greater than 0.2°C.

### Plasmids

Lentiviral packaging plasmid psPAX2 encoding the Gag, Pol, Rev, and Tat genes and pMDG2, which encodes VSV-G envelope, was obtained from Didier Trono (Addgene plasmids # 12260 and # 12259). HIV-1 Gag-Pol expression plasmid pCRV-1 (*Zennou et al., 2004*) and HIV-GFP encoding plasmid CSGW (*Naldini et al., 1996*) were kind gifts from Stuart Neil. The proviral construct pNL4.3-iGFP-ΔEnv was generated as described previously (*Aggarwal et al., 2012*). It contains the open-reading frame for eGFP flanked by protease cleavage sites inserted into the Gag gene between the coding sequences for MA and CA. In addition the start codon of the Env gene is mutated to a stop codon to prevent expression of the envelope protein.

### Preparation of HIV-1 virions

Replication deficient VSV-G pseudotyped HIV-1 virions were produced in HEK293T cells using pMDG2, pCRV GagPol and CSGW as described previously (*Price et al., 2014*). Viral supernatant from HEK293T cells was pelleted over a 20% sucrose cushion in a SW28 rotor (Beckman) at 28,000 rpm at 4°C. Pellets were resuspended in PBS. For removal of microvesicle contaminants, samples were treated with subtilisin essentially as described by *Ott (2009)*, with virions subsequently purified by ultracentrifugation through 20% sucrose. Subtilisin efficiency was determined by SDS PAGE and infection assay.

### Preparation of HIV-1 cores

HIV-1 capsid cores were prepared using a protocol based on *Shah and Aiken (2011)*. 90 ml HEK293T supernatant containing VSV-G pseudotyped HIV-1 GFP was pelleted over 20% sucrose dissolved in core prep buffer (CPB; 20 mM Tris (pH 7.4), 20 mM NaCl, 1 mM MgCl2) in an SW28 rotor (Beckman) at 25,000 rpm at 4°C. Pellets were gently resuspended at 4°C in CPB for 1 hr with occasional agitation. Resuspended pellets were treated with DNase I from bovine pancreas (Sigma Aldrich) for 1 hr at 200 µg/ml at room temperature to remove contaminating extra-viral DNA. Virus was subjected to spin-through detergent stripping of the viral membrane as follows. A gradient at 80–30% sucrose was prepared in SW40Ti ultracentrifuge tubes and overlaid with 250 µl 1% Triton X-100 in 15% sucrose, followed by 250 µl 7.5% sucrose. All solutions were prepared in CPB. 750 µl DNase-treated, concentrated virus was layered on top of the gradient and subjected to 32,500 rpm at 4°C for 16 hr. The preparation was fractionated and the location of cores was determined by ELISA for p24 (Perkin Elmer). Core-containing fractions were pooled and snap frozen before storage at −80°C.

## Preparation of cores for EM imaging

Cores were prepared as above. Pooled core fractions were diluted to reduce the sucrose concentration to 20%, then samples were spun 2 hr at 4°C at 45 K rpm in TLA55 rotor. Cores were resuspended in 8 µl CPB with 20% sucrose and loaded onto glow discharged carbon grids (Cu, 400 mesh, Electron Microscopy Services) for 5 min then stained for 3 min with 2% uranyl acetate. Micrographs were taken at room temperature on a Tecnai Spirit (FEI) operated at an accelerated voltage of 120 keV and Gatan 2k × 2 k CCD camera. Images were collected with a total dose of 30 e⁻/A °², between a defocus of 1–3 µm and a magnified pixel size of 0.93 nm/pixel.

## Encapsidated reverse transcription assays

Viral cores were diluted to 50 ng/ml p24 with 60% sucrose in CPB. Final concentrations of dNTPs were 1 µM each (unless otherwise indicated), DNase I and RNase A were at 100 mg/ml. 20 µl reactions were incubated at room temperature for 16 hr unless indicated otherwise. 4 µl of 5xMicrolysis Plus (Microzone) was added to each sample and processed according to manufacturers instructions. Reverse transcript products were detected using TaqMan Fast Universal PCR Mix (ABI) and RU5 primers to detect strong-stop DNA40 (RU5 forward: 5'-TCTGGCTAACTAGGGAACCCA-3'; RU5 reverse: 5'-CTGACTAAAAGGGTCTGAGG-3'; and RU5 probe 5'-(FAM) TTAAGCCTCAATAAAGC TTGCCTTGAGTGC(TAMRA)−3'), GFP primers to detect first-strand transfer products (described above) and primers for second-strand transfer products40 (2 ST forward: 5'-TTTTAGTCAGTG TGGAAAATCTGTAGC-3'; 2 ST reverse: 5'-TACTCACCAGTCGCCGCC-3'; and 2 ST probe: 5'-(FAM) TCGACGCAGGACTCGGCTTGCT(TAMRA)−3'). Unless otherwise stated, ERT data is representative of at least three independent experiments.

## Crystallisation, structure solution and analysis

All crystals were grown at 17°C by sitting-drop vapour diffusion in which 100 nl protein was mixed with 100 nl precipitant and suspended above 80 µl precipitant. The structures were all obtained from 10 to 12 mg/ml protein mixed with PEG550MME (13–14%), KSCN (0.15 M), Tris (0.1 M, pH 8.5) and cryoprotected with precipitant supplemented with 20% MPD. For the ATP-bound structure, the protein was supplemented with 1 mM ATP or IP$_6$ immediately before crystallisation. All crystals were flash-cooled in liquid nitrogen and data collected either in-house using Cu *Ka* X-rays produced by a Rigaku FR-E rotating anode generator with diffractionrecorded on a mar345 image plate detector (marXperts), or at beamline I02 at Diamond Light Source. The data sets were processed using the CCP4 Program suite (*Winn, 2003*). Data were indexed and integrated with iMOSFLM and scaled and merged with AIMLESS (*Evans and Murshudov, 2013*). Structures were solved by molecular replacement using PHASER (*McCoy, 2007*) and refined using REFMAC5 (*Murshudov et al., 1997*). Between rounds of refinement, the model was manually checked and corrected against the corresponding electron-density maps in COOT (*Emsley and Cowtan, 2004*). Solvent molecules and bound ligands were added as the refinement progressed either manually or automatically within COOT, and were routinely checked for correct stereochemistry, for sufficient supporting density above a $2Fo − Fc$ threshold of 1.0 s and for a reasonable thermal factor. The quality of the model was regularly checked for steric clashes, incorrect stereochemistry and rotamer outliers using MOLPROBITY (*Chen et al., 2015*). Final figures were rendered in The PyMOL Molecular Graphics System, Version 1.5.0.4 Schrödinger, LLC. Structures and data were deposited in the PDB database with codes 6ERM (ATP complex), 6ERN (AZT complex) and 6ES8 (IP6 complex).

## Single molecule measurements

Viral particles were produced by transfecting HEK293T cells with a mixture of pNL4.3-iGFP-ΔEnv and psPAX2 (1.4:1, mol/mol), collected 48 hr post transfection and viral membrane proteins were biotinylated using EZ-Link Sulfo-NHS-LC-LC-Biotin (Thermo Scientific). Biotinylated viral particles were purified by size exclusion chromatography using a HiPrep 16/60 Sephacryl S-500 HR column (GE Healthcare) and captured on the surface of a glass coverslip modified with PLL(20)-g[3.4]-PEG (2)/PEG(3.4)-biotin (Susos AG) and streptavidin. The viral envelope was permeabilised by addition of perfringolysin O (200 nM) in imaging buffer (50 mM HEPES pH 7.0, 100 mM NaCl) via microfluidic delivery and the diffraction-limited signal from the GFP-loaded viral particles was monitored by time-lapse total internal reflection fluorescence microscopy. Images were analysed with software

written in MATLAB (The MathWorks, Inc.). Fluorescence intensity traces were calculated for each viral particle in the field of view by integrating the fluorescence intensity in a $7 \times 7$ pixel region. Membrane permeabilisation was detected as a rapid drop in the signal resulting from the release of GFP contained in the viral particle in the volume outside the capsid (not shown). This step mimics viral fusion in that the capsid is exposed to the surrounding medium and was defined as time zero for measuring capsid opening. Particles with intact cores were identified as those with residual GFP signal after permeabilisation, arising from GFP molecules trapped inside the closed capsid. The onset of capsid uncoating was then detected as the sudden release (loss of the residual signal occurring typically within one frame) of the encapsidated GFP molecules via a sufficiently large defect in the capsid lattice.

## Production of $^3$H-inositol labelled virus

One $\times 10^6$ 293 T cells were seeded into $2 \times 10$ cm dishes in inositol-free DMEM and left to adhere overnight. The media was replaced with 5 ml inositol-free DMEM supplemented with 5 µCi/ml $^3$H-inositol (Perkin Elmer). After 3 days incubation, an additional 5 ml inositol-free media containing 5 µCi/ml $^3$H-inositol was added onto cells, which were then transfected with pMDG2, pCRV GagPol and CSGW. Cells were left for a further 3 days to produce VSV-G pseudotyped HIV1. Viral supernatants were topped up to 30 ml and pelleted over a 5 ml 20% sucrose cushion) in a SW28 rotor (Beckman) at 28,000 rpm at 4°C. Pellets were resuspended in inositol-free media and pelleted as previously. After the second spin, pellets were resuspended in 1 ml PBS and spun at 13,000 rpm at 4°C in a bench top microfuge for 60 min. Pellets were frozen at −20°C until processing. Cells were washed with PBS, then harvested by scraping, counted and pelleted for quantification of cellular IP$_6$ labelling. Pellets were frozen at −20°C until processing. For comparison of virion and purified capsid core samples, p24 levels were determined by ELISA for p24 (Perkin Elmer).

## Cells

293T CRL-3216 cells were purchased from ATCC and authenticated by the supplier. All cells are regularly tested and are mycoplasma free.

## Purification and HPLC analysis of inositol phosphates

Inositol phosphates extraction and analysis was performed modifying a previously described protocol (*Azevedo and Saiardi, 2006*). Cells and viral pellets were resuspended in 200 µl of extraction buffer (1M Perchloric acid, 5 mM EDTA) and incubated on ice for 10 min. The samples were spun out at 13,000 rpm at 4°C for 5 min and the supernatant recovered. Viral pellet were re-extracted for 10 min at 100°C using another 200 µl of extraction buffer and spun out as before. Supernatants from acid extractions were neutralised to pH6-8 using 1M Potassium carbonate, 5 mM EDTA (approximately 100 µl) and incubated on ice with the lids open for 1–2 hr. Samples were spun at 13,000 rpm at 4°C for 5 min and supernatant containing inositol phosphates was loaded onto HPLC or stored at 4°C. Inositol phosphates were resolved by strong anion exchange chromatography Sax-HPLC on a Partisphere SAX $4.6 \times 125$ mm column (Hichrom). The column was eluted with a gradient generated by mixing buffer A (1 mM EDTA) and buffer B (1 mM EDTA; 1.3 M $(NH_4)_2HPO_4$, pH 4.0) as follows: 0–5 min, 0% B; 5–10 min, 0–30% B; 10–85 min, 30–100% B; 85–95 min, 100% B. Fractions (1 ml) were collected and analysed by scintillation counting after adding 4 ml of Ultima-Flo AP LCS-cocktail (Perkin Elmer).

## Calculation of ATP and IP$_6$ per virion

The presence of ATP in virions was determined using a luciferase-based detection system following manufacturers instructions (Abcam ab113849). A sensitivity of 1 nM ATP detection was confirmed using a titration of 293 T cell lysate. The concentration of ATP estimated in 293 T cells assuming a volume of $1.77 \times 10^{-12}$ L was ~8 mM. The concentration of ATP estimated in HIV virions, assuming a volume of $5.2 \times 10^{-19}$ L, was ~1.5 mM. The level of IP$_6$ in 293 T cells was determined by PAGE analysis of TiO$_2$ purified inositol phosphates alongside polyphosphate standards (Sigma Aldrich S6128) and known standards of IP$_6$ (*Wilson et al., 2015*). For $4 \times 10^6$ cells there is estimated to be approximately 0.75 nmol IP$_6$. The calculated figure of 309 ± 41 IP$_6$ per virion is an average of three independent biological replicates.

## Acknowledgements

The authors wish to thank John Briggs for advice with EM and Miranda Wilson for 293T IP$_6$ measurements. LCJ is funded by the Medical Research Council (UK; U105181010) and through a Wellcome Trust Investigator Award. TB received funding from the Australian Centre for HIV and Hepatitis Virology Research and the National Health and Medical Research Council of Australia (APP1100771). GJT is funded by a Wellcome Trust Senior Fellowship, an ERC Advanced Grant (339223), the European Research Council under the European Union's Seventh Framework Programme (FP7/2007-2013) and the National Institute for Health Research. WM is supported by a Sir Henry Dale Fellowship jointly funded by the Wellcome Trust and the Royal Society (Grant Number 206248/Z/17/Z). DAJ is supported by an NHMRC Early Career Research Fellowship (CJ Martin) (GNT1036521). AS is supported by MRC core support to the MRC/UCL Laboratory for Molecular Cell Biology University Unit (MC_UU_1201814). CFD was supported by an NHMRC Early Career Fellowship (APP1110116).

## Additional information

### Funding

| Funder | Grant reference number | Author |
|---|---|---|
| Medical Research Council | U105181010 | Leo C James |
| Wellcome | | Gregory J Towers<br>Leo C James |
| National Health and Medical Research Council | 339223 | Till Böcking |
| National Health and Medical Research Council | GNT1036521 | David A Jacques |
| Wellcome | 206248/Z/17/Z | William A McEwan |
| National Health and Medical Research Council | APP1110116 | Claire F Dickson |

The funders had no role in study design, data collection and interpretation, or the decision to submit the work for publication.

### Author contributions

Donna L Mallery, Till Böcking, Conceptualization, Investigation, Writing—review and editing; Chantal L Márquez, Madhanagopal Anandapadamanaban, Katsiaryna Bichel, Adolfo Saiardi, Investigation; William A McEwan, Methodology; Claire F Dickson, Formal analysis; David A Jacques, Gregory J Towers, Writing—review and editing; Leo C James, Conceptualization, Supervision, Investigation, Writing—original draft, Project administration, Writing—review and editing

### Author ORCIDs

William A McEwan http://orcid.org/0000-0002-4408-0407
David A Jacques http://orcid.org/0000-0002-6426-4510
Till Böcking https://orcid.org/0000-0003-1165-3122
Leo C James http://orcid.org/0000-0003-2131-0334

### Decision letter and Author response

Decision letter https://doi.org/10.7554/eLife.35335.014
Author response https://doi.org/10.7554/eLife.35335.015

## Additional files

### Supplementary files

• Transparent reporting form
DOI: https://doi.org/10.7554/eLife.35335.012

## Data availability

Diffraction data have been deposited in PDB under the accession code 6ERM, 6ERN and 6ES8.

The following datasets were generated:

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
