## [Decision Letter]

Thank you for submitting your article "IP_6_ is an HIV pocket factor that prevents capsid collapse and promotes DNA synthesis" for consideration by *eLife*. Your article has been reviewed by two peer reviewers, and the evaluation has been overseen by Wes Sundquist as the Reviewing Editor and Wenhui Li as the Senior Editor. The following individual involved in review of your submission has agreed to reveal their identity: Hans-Georg Kräusslich.

The reviewers have discussed the reviews with one another and the Reviewing Editor has drafted this decision to help you prepare a revised submission.

Mallery et al. use a combination of biochemical and biophysical experiments to demonstrate that small polyanionic small molecules (e.g. (d)NTPs and, in particular, inositol hexakisphosphate (IP_6_)) bind within a positively charged ring in each CA hexamers that comprise the capsid lattice. The authors demonstrate that the functional consequence of these capsid-small molecule interactions is to stabilize the hexamer and consequently the mature capsid lattice in a dose-dependent fashion. This increase in core stability favors encapsidated reverse transcription (ERT), and the authors demonstrate that the observed increase in ERT is a consequence of increased capsid stability, with no secondary effects on the reverse transcriptase enzyme itself. The effects are most pronounced with IP_6_ – a molecule previously shown to promote the formation of native-like immature HIV-1 Gag particles in vitro. The authors find that IP_6_, as well as the IP4 and IP5 analogs, can stabilize capsid hexamers and promote ERT to varying degrees. Only IP_6_, however, is shown to be incorporated into virions, and the authors show directly that IP_6_ associates with the capsid, consistent with their biochemical and structural analyses.

Overall, this is an interesting and potentially important manuscript that identifies IP_6_ as a capsid stabilization factor and sheds light on HIV-1 capsid stability and uncoating. The convincing demonstration that IP_6_ binds the capsid hexamer pore and stabilizes the capsid at low concentrations, thereby enhancing endogenous reverse transcription is an important one, with significant implications for the HIV post-entry phase, although there remain some questions about the different role(s) of the pore in binding and passing different poly-charged anions (e.g., dNTPs vs. ATP vs. IP_6_).

Some issues remain to be addressed before publication.

Significant issues that need to be addressed:

1) The authors have not carefully documented the purity of their virus preparations, and it is well known that cells produce microvesicles of similar size and density as HIV-1. Though Figure 7C includes a control (please indicate exactly what this is: one assumes supernatant of uninfected cells, though this information seems to be lacking), HIV-1 infection could stimulate microvesicle production. It therefore is important to demonstrate virus prep purity experimentally. For example, the authors should include subtilisin treatment in at least one of the relevant experiments (if this has not already been done). The subtilisin protease treatment protocol was developed to destroy contaminating microvesicles, and therefore is a gold standard for addressing virion incorporation (and of course it also digests surface viral proteins, but leaves the virions and core proteins intact).

The issue of virion purity is of particular importance in the report of levels of IP_6_ in the virion. This is an important number and should not be published without better evidence for virion purity.

2) ATP and non-hydrolysable analogs are also shown to stabilize the CA hexamer and stimulate ERT, though in Figure 4D authors importantly show that IP_6_ stimulates at much lower concentrations than ATP. However, if ATP is present in cells at much higher concentration, could it be that ATP, not IP_6_, is the more relevant capsid stabilization factor and pocket binding factor? Is ATP incorporated into virus? If so, at what level (an absolute number isn't essential, but it would be useful to understand whether ATP can be a relevant pocket factor)?

3) In Jacques 2016, the authors concluded that R18 residues at the CA hexamer vertex bind dNTPs and act as pores that selectively incorporate RT dNTP substrates into the capsid. These results met with some skepticism because, prior to the paper, the field assumed that small molecules such as drugs or NTPs freely diffuse, while larger molecules such as proteins required some degree of CA uncoating to access the core. The authors now importantly show that NNRTIs do not interact with the R18 CA pore, and suggest that "NNRTIs are sufficiently small to pass through the pore and so may enter by simple diffusion." This statement seems puzzling in light of the fact that RVP (studied here), is 366 daltons, dCTP is 467 Da and the NNRTI etavirine (not studied here), is 435 Da. It seems hard to believe that the difference between 467 and 435 d dictates active import vs. passive diffusion. It now seems more likely that the R18 pore is a promiscuous docking site for molecules of appropriate sized and charge, and that the primary biological role of such interactions is to regulate capsid stability (as implied in the current study), rather than to mediate selective transport. The authors should discuss these issues explicitly and explain their current thinking.

4) Given the focus on capsid stability and the fact that the HIV-1 capsid has a very distinctive morphology, it is surprising that the authors have not included any EM images in their report. While detailed analyses of core completeness requires cryo-electron tomography (and is therefore probably beyond the current studies), good indications of core integrity, purity, and alterations with time should be visualized using negative stain EM.

5) The authors generally use +/-DNase treatment in their ERT reactions and show that DNase treatment is required to see clear effects (consistent with the idea that RT is occurring within intact capsids). In the absence of DNase treatment, greater levels of "background" DNA synthesis are observed and therefore the effects of capsid stabilizers are relatively minor in the absence of DNase. Why then is the experiment in Figure 5A done without DNase, and why isn't "background" RT observed in this case?

Other issues for the author's consideration:

1) In the subsection “Inositol phosphate IP_6_ stabilises capsid hexamers and promotes DNA accumulation during RT”, the conclusion 'to a significantly different degree' would be bolstered by including ATP in the experiment.

2) The Gag polyprotein should be written with a capital G.

3) Descriptions of reagents and methods are often very terse and should be expanded to the point where others could repeat the experiments. As one example, it will not be immediately clear to readers that a Cys-mutated CA protein is being used in some of the experiments. Similarly, the prose is often quite terse (as one example, it would be easier on the reader if a bit more explanation were provided with short statements like "IP_6_ had no effect on monomeric capsid').

4) Data obtained using the same experimental methods are represented very differently in different figures without making it clear why this may be needed. The manuscript would be more accessible for non-experts if more attention were paid to this issue.

5) The statement "IP_6_ potently stabilized capsid cores and to a much greater extent than ATP" should be rephrased since at 1 mM both factors had the same effect (it's really the concentration range that's different).

6) In the subsection “Inositol phosphate IP_6_ stabilises capsid hexamers and promotes DNA accumulation during RT”, "specifically" seems unwarranted, as ATP stimulates basically the same level of ERT product formation.

7) In the subsection “NNRTIs inhibit encapsidated RT but don’t bind capsid pores”: The RVP response looks significantly different vs buffer.

8) Plasmids are listed at different sites in the Materials and methods (subsections “Single molecule measurements” and “Production of ^3^H-inositol labelled virus”) and without appropriate citations. Please consider adding short "plasmids" section to the Materials and methods for clarity.

9) In the subsection “Calculation of IP_6_ per virion”. A supplementary figure is cited, but no supplementary information was included in the reviewer's version of the manuscript.

10) What is Figure 7F marker lane?

11) In the subsection “Inositol phosphate IP_6_ stabilises capsid hexamers and promotes DNA accumulation during RT”, "hexakisphosphate" is misspelled.

12) In the subsection “Inositol phosphate IP_6_ stabilises capsid hexamers and promotes DNA accumulation during RT”. The stoichiometry claim is a bit unclear. The data in Figure 4B clearly show approximately half maximal binding at 20 µM. The Materials and methods state that "hexamer samples were prepared at a final concentration of 200 µM". It appears that the concentration of CA was reported relative to monomeric CA, but 200 µM CA monomer is ~33 µM CA hexamer, and this value makes more sense given the data.

13) Please clarify what is meant by the "abundant inositol fraction" being eluted during Sax-HPLC?

14) Please check abbreviation use carefully. ERT is defined early on as "encapsidated reverse transcription", but the spelled out term then reappears numerous times. Then in the Materials and methods, the term "endogenous reverse transcription" is used (which is actually the more common ERT definition). NNRTI comes in without definition, etc.

Additional statistical issues:

1) The authors make rather sweeping statements about data significance without providing appropriate analyses. e.g., "significant ATP stimulation was only observed in reactions carried out in the presence of nuclease". The initial two bars in Figure 2D would likely indicate significance if proper statistical test (e.g., t test) were performed, but statistical tests should be applied comprehensively and the text amended as required.

2) Although some datasets have error bars, numbers of biological replicates (independent experiments) vs. technical replicates (number of replicates within individual experiments) are often not well described. An easy fix would be to state this info in figure legends. Moreover, it is not always clear that multiple biological replicates were actually performed in all cases (e.g., in the Materials and methods, the authors indicate that DSF scans are single reads, but individual DSF experiments should show to be reproducible in independent runs). Similarly, please indicate the number of biological replicates for representative Figure 1A, Figure 3B, Figure 4B, Figure 7A data.

3) Data shown in Figure 2B, Figure 3B, Figure 7D and 7E should also apparently have error bars. Also, please define all error bars (SEM, SD, etc.).

---

## [Author Response]

Significant issues that need to be addressed:1) The authors have not carefully documented the purity of their virus preparations, and it is well known that cells produce microvesicles of similar size and density as HIV-1. Though Figure 7C includes a control (please indicate exactly what this is: one assumes supernatant of uninfected cells, though this information seems to be lacking), HIV-1 infection could stimulate microvesicle production. It therefore is important to demonstrate virus prep purity experimentally. For example, the authors should include subtilisin treatment in at least one of the relevant experiments (if this has not already been done). The subtilisin protease treatment protocol was developed to destroy contaminating microvesicles, and therefore is a gold standard for addressing virion incorporation (and of course it also digests surface viral proteins, but leaves the virions and core proteins intact).The issue of virion purity is of particular importance in the report of levels of IP_6_ in the virion. This is an important number and should not be published without better evidence for virion purity.

The control in Figure 7C is as assumed above and we have introduced a clarifying sentence into the legend to make this clear. We are grateful for the suggestion to use subtilisin treatment as a way of improving viral prep purity and addressing the issue of microvesicle contamination. In our revised manuscript, we present data on subtilisin treated virions, following the established protocol by Ott. Importantly, we verified that treatment reduced contamination by SDS PAGE and that it acted on membrane associated protein by performing infection experiments. This new data is collated in Extended Data Figure 7. We agree that the issue of virion purity is most important to our IP_6_ incorporation data and have therefore repeated our tritiated inositol experiments using subtilisin treated virions. This new data has been added as Figure 7D and shows that measured IP_6_ is present in samples treated to remove contaminating microvesicles. This is consistent with the fact that we measure little IP5 in samples despite the fact that it is present in cells at concentrations ~ 25% those of IP_6_ and supports the notion that IP_6_ is selectively enriched within virions. The new subtilisin data also agrees very well with the data in Figure 7F, which shows that the IP_6_ in virions is associated with the capsid core. This data was obtained measuring the IP_6_ in purified capsids cores that have been stripped of membrane contaminants using triton. Taken together we believe this provides excellent evidence that IP_6_ is associated with the capsid in viral particles.

2) ATP and non-hydrolysable analogs are also shown to stabilize the CA hexamer and stimulate ERT, though in Figure 4D authors importantly show that IP_6_ stimulates at much lower concentrations than ATP. However, if ATP is present in cells at much higher concentration, could it be that ATP, not IP_6_, is the more relevant capsid stabilization factor and pocket binding factor? Is ATP incorporated into virus? If so, at what level (an absolute number isn't essential, but it would be useful to understand whether ATP can be a relevant pocket factor)?

We are grateful for this suggestion and have now tested for ATP incorporation. Using a luciferase assay that is sensitive to concentrations of 1 nM ATP, we show that ATP is present in HIV-1 virions. An approximate calculation suggests that the concentration in virions is similar to that in the producer 293T cells, which we measured in the same assay. This new data has been incorporated in Figure 2G.

We agree that it is possible that ATP is a capsid stabilization factor and to acknowledge this more clearly in the manuscript we have inserted the following into the Discussion:

“While we propose that IP_6_ serves as an HIV pocket factor we cannot rule out that other cellular polyanions also play a role. For instance, although we observed no evidence for enrichment, ATP is highly abundant and was readily detected in virions.”

3) In Jacques 2016, the authors concluded that R18 residues at the CA hexamer vertex bind dNTPs and act as pores that selectively incorporate RT dNTP substrates into the capsid. These results met with some skepticism because, prior to the paper, the field assumed that small molecules such as drugs or NTPs freely diffuse, while larger molecules such as proteins required some degree of CA uncoating to access the core. The authors now importantly show that NNRTIs do not interact with the R18 CA pore, and suggest that "NNRTIs are sufficiently small to pass through the pore and so may enter by simple diffusion." This statement seems puzzling in light of the fact that RVP (studied here), is 366 daltons, dCTP is 467 Da and the NNRTI etavirine (not studied here), is 435 Da. It seems hard to believe that the difference between 467 and 435 d dictates active import vs. passive diffusion. It now seems more likely that the R18 pore is a promiscuous docking site for molecules of appropriate sized and charge, and that the primary biological role of such interactions is to regulate capsid stability (as implied in the current study), rather than to mediate selective transport. The authors should discuss these issues explicitly and explain their current thinking.

We have inserted the following section into the Discussion:

“In our original discovery of the R18 pore and its importance in HIV reverse transcription and infection we postulated that it provided a channel to import nucleotides to drive ERT. […] Either way, this indicates that small molecules not bound by the R18 ring can still enter the capsid through the pore, or through solvent channels as has previously been proposed.”

4) Given the focus on capsid stability and the fact that the HIV-1 capsid has a very distinctive morphology, it is surprising that the authors have not included any EM images in their report. While detailed analyses of core completeness requires cryo-electron tomography (and is therefore probably beyond the current studies), good indications of core integrity, purity, and alterations with time should be visualized using negative stain EM.

We did not include electron microscopy in our original study as it is not a method we have expertise in and because we felt that our single molecule approach would allow better quantification of capsid stability in a time and condition dependent manner. However, with the help of experienced colleagues in our institute we have now carried out negative stain experiments to visualize cores in our samples. Importantly, these EM experiments were carried out using capsid cores that were prepared using the same triton spin-through method we use to generate material for our encapsidated reverse transcription (ERT) and IP_6_ incorporation assays. These new data are presented in Figure 2A and show that there are intact cores with the expected morphology in our samples. In order to concentrate samples sufficiently for visualization, cores were pelleted by centrifugation. This process does damage cores to some degree as we observe a loss of efficiency in ERT when using pelleted samples. We also observed considerable variation between grids and have therefore not attempted to use this approach to quantify capsid integrity under different conditions and timepoints as we have done using our single molecule experiments. Nevertheless, this new EM data demonstrates the presence of capsid cores with the expected morphology in our preparations.

5) The authors generally use +/-DNase treatment in their ERT reactions and show that DNase treatment is required to see clear effects (consistent with the idea that RT is occurring within intact capsids). In the absence of DNase treatment, greater levels of "background" DNA synthesis are observed and therefore the effects of capsid stabilizers are relatively minor in the absence of DNase. Why then is the experiment in Figure 5A done without DNase, and why isn't "background" RT observed in this case?

Experiment 5A was carried out without DNase in order to test the stabilizing effect of IP_6_*prior* to reverse transcription (RT). Nuclease is required to remove RT products from outside the virion once they have formed, whereas we reasoned that capsids that have fallen apart prior to addition of nucleotides will not support DNA synthesis because RT enzyme and template will be diluted into bulk solvent. This is supported by the data, which shows that cores incubated for 24hrs prior to nucleotide addition are no longer capable of RT whereas cores that are given IP_6_ are. As nuclease was not required for the experiment it was not necessary to add it, moreover the resulting data provided a measurement of core stability that was not dependent upon nuclease addition. Background RT is observed in Figure 5A and is similar to the background in our other experiments (e.g. Figure 4C).

Other issues for the author's consideration:1) In the subsection “Inositol phosphate IP6 stabilises capsid hexamers and promotes DNA accumulation during RT”, the conclusion 'to a significantly different degree' would be bolstered by including ATP in the experiment.

We have performed a side-by-side comparison and included this as a new figure panel (Figure 4C).

2) The Gag polyprotein should be written with a capital G.

Corrected.

3) Descriptions of reagents and methods are often very terse and should be expanded to the point where others could repeat the experiments. As one example, it will not be immediately clear to readers that a Cys-mutated CA protein is being used in some of the experiments. Similarly, the prose is often quite terse (as one example, it would be easier on the reader if a bit more explanation were provided with short statements like "IP6 had no effect on monomeric capsid').

We have removed the sentence and clarified the first point:

“As previously, we used a mutant hexamer protein that is coordinated by disulphide bridges between monomers through the introduction of mutations A14C and E45C and with additional dimerization mutants W184A and M185A.”

4) Data obtained using the same experimental methods are represented very differently in different figures without making it clear why this may be needed. The manuscript would be more accessible for non-experts if more attention were paid to this issue.

We have clarified as follows:

“We therefore tested by DSF whether IP_6_ can stabilise the mature capsid and promote ERT by facilitating the accumulation of RT products within the core. […] To show the transition between stabilised species, the melt curves rather than change in stability (ΔT_m_) are shown (Figure 4B).”

5) The statement "IP6 potently stabilized capsid cores and to a much greater extent than ATP" should be rephrased since at 1 mM both factors had the same effect (it's really the concentration range that's different).

The statement has been rephrased to “IP_6_ potently stabilised capsid cores at much lower concentrations than ATP.”

6) In the subsection “Inositol phosphate IP6 stabilises capsid hexamers and promotes DNA accumulation during RT”, "specifically" seems unwarranted, as ATP stimulates basically the same level of ERT product formation.

“Specifically” has been removed.

7) In the subsection “NNRTIs inhibit encapsidated RT but don’t bind capsid pores”: The RVP response looks significantly different vs buffer.

We have performed multiple repeats of this experiment in the presence and absence of DTT and find that while there is an RVP response it is independent of hexamer assembly. We have included this data in the revised manuscript (Figure 3A) and introduced the following into the text:

“We first used DSF to test interaction with capsid hexamers and found that only AZT and not the hydrophobic compounds RVP or NVP matched the stabilisation of dATP (Figure 3A). […] This suggests any RVP interaction is not due to the R18 hexameric ring as this feature will be destroyed upon DTT addition because the disulphides holding the hexamer together become reduced.”

8) Plasmids are listed at different sites in the Materials and methods (subsections “Single molecule measurements” and “Production of ^3^H-inositol labelled virus”) and without appropriate citations. Please consider adding short "plasmids" section to the Materials and methods for clarity.

A separate plasmids section has been added to the Materials and methods as requested.

9) In the subsection “Calculation of IP6 per virion”. A supplementary figure is cited, but no supplementary information was included in the reviewer's version of the manuscript.

This has been removed.

10) What is Figure 7F marker lane?

The marker lane contains polyphosphate standards obtained from Sigma-Aldrich. This information has now been included in the Materials and methods and relevant figure legend.

11) In the subsection “Inositol phosphate IP6 stabilises capsid hexamers and promotes DNA accumulation during RT”, "hexakisphosphate" is misspelled.

Corrected. Thank you!

12) In the subsection “Inositol phosphate IP6 stabilises capsid hexamers and promotes DNA accumulation during RT”. The stoichiometry claim is a bit unclear. The data in Figure 4B clearly show approximately half maximal binding at 20 µM. The Materials and methods state that "hexamer samples were prepared at a final concentration of 200 µM". It appears that the concentration of CA was reported relative to monomeric CA, but 200 µM CA monomer is ~33 µM CA hexamer, and this value makes more sense given the data.

We have clarified the concentrations in the Materials and methods and introduced the following into the Results:

“A concentration of 20 µM IP_6_ gave approximately half maximal stabilisation to a solution of 33 µM hexamer. This suggests that hexamer stabilisation by IP_6_ is likely stoichiometric, with one IP_6_ molecule per hexamer.”

13) Please clarify what is meant by the "abundant inositol fraction" being eluted during Sax-HPLC?

We have inserted into the Results the following replacement text:

“To confirm that our core prep has removed enveloped particles we also compared the relative levels of inositol. […] Consistent with this, we observed loss of the inositol peak in the capsid only sample (Figure 7E).”

14) Please check abbreviation use carefully. ERT is defined early on as "encapsidated reverse transcription", but the spelled out term then reappears numerous times. Then in the Materials and methods, the term "endogenous reverse transcription" is used (which is actually the more common ERT definition). NNRTI comes in without definition, etc.

Abbreviations have been corrected.

Additional statistical issues:1) The authors make rather sweeping statements about data significance without providing appropriate analyses. e.g., "significant ATP stimulation was only observed in reactions carried out in the presence of nuclease". The initial two bars in Figure 2D would likely indicate significance if proper statistical test (e.g., t test) were performed, but statistical tests should be applied comprehensively and the text amended as required.

The term ‘significant’ has been removed where statistical analysis has not been performed.

2) Although some datasets have error bars, numbers of biological replicates (independent experiments) vs. technical replicates (number of replicates within individual experiments) are often not well described. An easy fix would be to state this info in figure legends. Moreover, it is not always clear that multiple biological replicates were actually performed in all cases (e.g., in the Materials and methods, the authors indicate that DSF scans are single reads, but individual DSF experiments should show to be reproducible in independent runs). Similarly, please indicate the number of biological replicates for representative Figure 1A, Figure 3B, Figure 4B, Figure 7A data.

Replicates are now indicated in figures as suggested. Additional information has also been given in the appropriate methods (e.g. for DSF and ERT).

3) Data shown in Figure 2B, Figure 3B, Figure 7D and 7E should also apparently have error bars. Also, please define all error bars (SEM, SD, etc.).

Error bars have now been defined where given. It has also now been stated where single measurements representative of independent experiments are shown.